# Biomechanical Analysis of Palateless Splinted and Unsplinted Maxillary Implant-Supported Overdentures: A Three-Dimensional Finite Element Analysis

**DOI:** 10.3390/ma16155248

**Published:** 2023-07-26

**Authors:** Mária Frolo, Luboš Řehounek, Aleš Jíra, Petr Pošta, Lukáš Hauer

**Affiliations:** 1Department of Stomatology, Faculty of Medicine in Pilsen, Charles University, 304 60 Pilsen, Czech Republic; postap@fnplzen.cz (P.P.); hauerl@fnplzen.cz (L.H.); 2Department of Mechanics, Faculty of Civil Engineering, Czech Technical University in Prague, 166 29 Prague, Czech Republic; lubos.rehounek@fsv.cvut.cz (L.Ř.); jira@fsv.cvut.cz (A.J.)

**Keywords:** FEA, implant-supported overdenture, locator attachments, bar-retained overdenture

## Abstract

The objective of this study was to compare the distribution of stress in the maxillary bone, dental implants, and prosthetic components supporting implant-supported maxillary overdentures with partial palatal coverage, in both splinted and unsplinted designs. Two models of maxillary overdentures were designed using the Exocad Dental CAD program, which included cancellous and cortical bone. The complete denture design and abutments (locator abutments in the unsplinted and Hader bar with Vertix attachments placed distally in the splinted variant) were also designed. The denture material was PEEK (Polyetheretherketone), and the method used to analyze patient-specific 3D X-ray scans was 3D QCT/FEA (three-dimensional quantitative computed tomography-based finite element analysis). Loading was divided into three load cases, in the frontal region (both incisors of the denture) and distal region (both molars and first premolar of the denture). The forces applied were 150 N with an oblique component with a buccal inclination of 35° in the frontal region, and 600 N with a buccal inclination of 5° (molars) or solely vertical (premolar) in the distal region. The model with locator abutments showed higher stresses in all load cases in both analyzed implant variants and in the maxilla. The differences in stress distribution between the splinted and unsplinted variants were more significant in the distal region. According to the results of the present study, the amount of stress in bone tissue and dental implant parts was smaller in the splinted, bar-retained variant. The findings of this study can be useful in selecting the appropriate prosthetic design for implant-supported maxillary overdentures with partial palatal coverage.

## 1. Introduction

Despite a decline in the overall rate of edentulousness in the past decade, demographic aging has made it remain a significant issue worldwide. In addition to traditional rehabilitation methods using total removable dentures, implant-supported solutions have been increasingly integrated since the official adoption of tooth loss rehabilitation with dental implants in 1978. Dental implants can be utilized in fixed restorations, or as a way to stabilize a total removable denture, such as in an implant-supported overdenture (IOD) [1].

IOD is a treatment option that requires proper assessment and appropriate management from the outset, with indications for its use first defined in the 1990s [2]. It is ideal for patients with insufficient bone quality, a buccal inclination of the residual alveolar ridge, thin and highly mobile mucosa, a high lip line, and a lack of lip support to achieve adequate phonetics and esthetics [3].

Attachment systems available to connect an overdenture to the supporting dental implants include unsplinted attachments (such as various ball types, magnets, locators, and telescopic double crowns) and splinted attachments (such as bars and clips in various designs) [4,5,6]. However, no consensus has yet been reached on which is better [7]. When planning therapy and selecting the attachment for an IOD, important factors to consider include cost efficiency, the required retention amount, the discomfort imposed on soft tissue, the amount of bone present, the expected level of oral hygiene, the patient’s social status, the patient’s expectations, the intermaxillary relationship, the state of opposing teeth, and the distance between implants [8].

Bar-retained restorations offer greater retention and show improved stability for dentures [9]. Bars are able to withstand lateral movement and rotation during function, allow for even distribution of force between fixtures, and correct potential implant axis discrepancies [6,10]. However, the use of bars may increase the likelihood of mucositis and mucosal hyperplasia due to poor oral hygiene, and there is a certain interocclusal space needed to meet the requirements of the denture [11].

While unsplinted construction may offer advantages in terms of treatment simplicity, duration, hygiene, and cost [12], the evidence regarding the impact of splinting versus not splinting on the load distribution of maxillary overdentures is still relatively scarce, and somewhat contradictory [13]. Although Cavallaro et al. suggested that unsplinted implants retaining maxillary overdentures with partial palatal coverage are viable, their observation period was only 48 months [14].

In a literature review with a similar mean follow-up period, Raghoebar et al. found an average survival rate of 88.9% for unsplinted implants in the maxilla, while splinted attachments had an implant survival rate of over 97% [15]. Conversely, a 2019 systematic review by DiFrancesco et al. found no statistical difference in maxillary overdentures supported by four implants between the splinted and unsplinted implant groups in terms of implant survival, overdenture durability, and patient satisfaction [16].

The lack of standardization in prosthetic techniques, implant numbers, measurements of marginal bone, and loading conditions in studies may account for the subjective preference for using unsplinted or splinted attachments for maxillary overdentures [2]. Therefore, there is still a need for a more accurate evaluation of the biomechanical differences between these two options.

The finite element method (FEM) is a numerical approach that aims to reduce complex problems to a system of algebraic equations, providing insight into the behavior of the system under consideration. The FEM is particularly useful in solving problems with intricate geometries and numerical solutions to extremely complicated stress problems may be routinely achieved using finite element analysis (FEA) [17]. The finite element analysis has been successfully applied to study stress and strain in implant dentistry [18]. With the increasing availability of 3D imaging techniques, such as quantitative computed tomography (QCT), it is now possible to reconstruct the maxillary bone accurately and perform QCT/FEA to analyze stress distribution at the bone–implant interface.

This study aimed to compare the stress distribution in maxillary bone, dental implants, and prosthetic components supporting implant-supported maxillary overdentures with partial palatal coverage in both splinted and unsplinted designs using 3D FEA. The null hypothesis was that the splinted design would exhibit superior biomechanical performance compared to the unsplinted design. To either prove or disprove this hypothesis, this study is therefore limited only to these two types of IODs, and aims to distinguish the differences between them.

While many FEA analyses assume that bone tissue is linearly elastic, isotropic, and homogeneous, these assumptions do not accurately reflect the complex nature of real bone tissue [19]. However, the use of 3D imaging and QCT/FEA can overcome this limitation and provide more realistic simulations of bone–implant interface behavior [20]. Therefore, the present study aims to provide more accurate and reliable insights into the biomechanical performance of implant-supported maxillary overdentures.

## 2. Materials and Methods

The present study was conducted at the Faculty of Civil Engineering, Czech Technical University in Prague, in cooperation with the Department of Stomatology, Faculty of Medicine in Pilsen, Charles University. The patient selected for this study was a 76-year-old male with heart surgery and a pacemaker in anamnesis and sufficient dexterity. Written consent was obtained from the attendee of the study.

Extraoral and intraoral requirements for an implant-supported overdenture in the maxilla were met [3]. The lip line was high (the patient had a significant gummy smile with visible teeth; wax-up try-in is depicted in Figure 1) and the lip support was essential. The buccal inclination of the residual ridge (Figure 2) was present, and the distance between the residual ridge and the incisal margin of lower incisors was more than 1.5 cm. Insufficient bone in distal regions with sufficient bone in the frontal region predetermined the patient for an anterior maxillary concept.

Cylindrical implants from Straumann (SLA RN SP Roxolid${}^{®}$) manufacturer have been implanted in the sites with the most generous bone offer as listed: the site of the upper-right first premolar (4.8 mm in diameter and 14 mm in length), upper-right first incisor (3.3 mm in diameter and 12 mm in length), upper-left second incisor (4.1 mm in diameter and 12 mm in length), and upper-left first premolar (4.1 mm in diameter and 12 mm in length). The 3D (three-dimensional) X-ray scans obtained with Planmeca ProMax${}^{®}$ 3D Classic show the location of the dental implants and the bone present (Figure 3).

The method used to analyze patient-specific 3D X-ray scans obtained with Planmeca ProMax${}^{®}$ 3D Classic using Romexis${}^{®}$ 3D imaging was QCT/FEA. The 3D X-ray, the position of the implants, their shape, and their location were modeled according to data provided by the patient. The analyzed implant variants can be seen in Figure 4.

The model included cancellous and cortical bone. The complete denture design and abutments (locator abutments in the unsplinted and Hader bar with Vertix attachments placed distally in the splinted variant) were designed using the Exocad Dental CAD (Exocad Gmbh Darmstadt, Germany) program in the dental laboratory. The denture was designed as palateless. The denture with splinted variant is depicted in Figure 5.

The denture material (including teeth) was PEEK (Polyetheretherketone). Material properties are presented in Table 1.

### 2.1. Methodology

The software used for the analyses was Mechanical Finder v12.0. (RCCM, Tokyo, Japan) [21]. The software performs QCT/FEA analyses on sliced image data, such as CT or roentgen scans. QCT/FEA enables fully inhomogeneous analyses where each voxel (or element in the FEM model) extracted from the range of interest in the sliced data has unique material properties based on the individual patient’s bone quality. This presents a great step up to other conventional methods of modeling, where bone is usually represented by a two-phase model (cancellous, cortical). In the present study, the patient’s maxilla was reconstructed from a roentgen scan, and 2 geometrical variants of implants were analyzed.

### 2.2. Mesh, Loads, and Boundary Conditions

The total number of nodes, shells, and solids of splinted and unsplinted models are listed in Table 2. In both modeled variants, loading was divided into 3 load cases, the first one being the frontal region (both incisors of the denture), with a loading force of 150 N, with the oblique component of the force set in the buccal direction with an inclination of 35°. The second and third ones were located in distal regions (both molars and the first left premolar). The simulated biting force applied on the molar had an oblique component with a much smaller buccal inclination than in the frontal region (5°), and the loading force was set to 600 N. The direction was solely vertical in the premolar region, with the loading force set to 600 N. To compare both models in the images containing the findings, the upper and lower limits of the color bar were normalized.

The material properties were exactly the same in both splinted and unsplinted alternatives. Thanks to the use of PEEK for the prosthesis fabrication, no metallic framework for either alternative was required. Both simulations were performed using a personal computer with 64 GB of DDR4 3200 MHz RAM and a 16-thread, 4.7 GHz processor.

## 3. Results

### 3.1. Comparison of Stress Distribution between Splinted and Unsplinted Models in Implants

When force was applied in the frontal region (both incisors of the denture, with a loading force of 150 N with the oblique component of the force set in the buccal direction), the stress distribution was similar in both variants, and the equivalent stress did not show many differences between the two variants. Even in the unsplinted variant, the load was well distributed between the implants due to the force distribution facilitated by the denture (Figure 6 and Figure 7).

However, when the load was applied in the molar region (oblique component with 5° inclination) and the loading force was set to 600 N, the splinted variant showed good resistance to flexure, whereas in the unsplinted variant, most of the load was localized to the outermost implant (Figure 8 and Figure 9).

With the force applied to the first premolar (direction solely vertical, with a loading force set to 600 N), the differences in stress distribution between the unsplinted and splinted variant were not significant. The load was transferred into the body of the implant positioned directly under the first premolar, so the effect of bending was diminished. Therefore, the bar only helped partially. The load distribution was also facilitated through the denture, which reduced the differences between the two variants (Figure 10 and Figure 11).

### 3.2. Comparison of Stress Distribution between Splinted and Unsplinted Models in the Maxilla

However, the differences in distributions of minimum principal stress, which was chosen for the evaluation of bone, were more severe. In the unsplinted variant, the stiff implants (which themselves resist the load quite well) were pushed into the bone, which resulted in peaks of compressive stress in the bone at the implants’ apices. This effect was more pronounced in the distal region when loading was applied to the premolar and molars of the denture (Figure 12).

In the splinted variant, the excessive implant displacement was prevented by the bars. The compressive stresses are noticeably smaller in all regions where the load was applied, and were localized in the region adjacent to the axis of the implant (Figure 13).

## 4. Discussion

According to the results of the present study, the amount of stress in bone tissue and dental implant parts was smaller in the splinted, bar-retained variant. The null hypothesis was therefore confirmed.

It is beyond question that masticatory forces are transferred to the dental restoration, and are not diminished but converted to energy distributed via the restoration–implant complex. Splinting enhances the functional support area and the anterior–posterior space that resists lateral loading and the retention of cement in fixed cemented restorations and reduces the potential for abutment screw dislodgement, loss of marginal bone surrounding the implants, and the likelihood of fracture of the prosthetic components [24]. Positive biomechanical results of splinting in fixed implant-supported rehabilitations have been proven many times [24,25,26].

For the removable implant-supported overdentures, there is a variety of results for the splinting effect. For mandibular implant-supported overdentures, Tabata et al. [27] found in their FEA study while comparing the effect of unsplinted and splinted implants using bars on the distribution of stress in mandibular implant-supported overdentures that implants connected with bar–clip attachments are favored over the O-ring group. A similar conclusion was achieved by Vafei et al. [28], where better stress–strain relationships were associated with the use of bar–clip attachments, especially in the protrusive motion, while for laterotrusive motion, the stress distribution was comparable with ball-supported overdentures. Jofre et al. [29] performed an FEA study together with a prospective clinical study to determine the splinting effect in the biomechanics of mini-implants. They stated that the minimum principal stress in the unsplinted group was more than twice as high as in the splinted group (−118.0 vs. −56.8 MPa), and after two years of follow-up, the ball attachment exhibited considerably higher overall prevalence of vertical bone loss. Assunção et al. [30] compared stress distribution in conventional complete dentures and IODs in the mandible, where the use of an attachment system increased stress values, which were higher when an axial abutment was used in comparison to a bar–clip attachment system.

By performing FEA, the study conducted by Barão et al. [31] came to the conclusion that two unsplinted implants with O-ring attachments showed the lowest stress value in comparison with bar attachments and also exhibited improved stress distribution. The findings were implicitly linked with the deformation of the mandible under loading, creating torsion in the centric portion of the mandible. The unsplinted design was able to adapt to the bone distortion without altering it, while the rigid bar had a tendency to counteract this movement, increasing the stress values. Menicucci et al. [32] also related the higher peri-implant bone stress with bar attachment to jaw deformation during mastication.

For maxillary overdentures, the results in the literature are scarce. A 3D FEA study conducted by Geramy et al. [33] corroborates the results of the present study. Similarly, Aquib et al. [34], who evaluated stress distribution using FEA in four implant models with ball attachments and implants connected with a bar, came to the conclusion that the bar–clip attachment showed the least stress in the maxilla. When the palatal coverage on implant-retained maxillary overdentures was investigated using FEA, the results were inconsistent. Kim et al. [35], in a study from 2016, suggested that full palatal coverage is more beneficial for stress distribution, while they preferred the usage of the Hader bar to the milled bar. On the other hand, Fernandez et al. [36] found no significant difference under axial loading between the prosthesis with full palatal coverage and without palatal coverage when the bar was added. Under nonaxial loading, the decrease in stress was observed with the bar attachment in all sites, with the exception of the anterior implant site.

In some literature reviews, the implant loss was higher with ball-retained than bar-supported overdentures [15,37]. In others, like the systematic review and meta-analysis conducted in 2018 [38], the results of splinted and unsplinted overdenture attachment systems achieved similar results. Also similar to the previous literature review was a literature review from 2019 [39], where no difference in clinical outcomes was found in studies concerning mandibular overdentures. In terms of implant survival, overdenture longevity, and patient satisfaction, Di Francesco et al. [16] concluded no statistical difference between splinted and unsplinted groups. Stoumpis et al. [40] concluded that no significant difference was found between splinted and unsplinted design in both maxilla and mandible in the peri-implant outcome, soft tissue health status, or patient satisfaction, even though bar-supported overdentures have been reported to require less prosthetic maintenance. Higher maintenance and repair requirements were also found in a review from 2020 [41] for locator attachments.

Based on the results of the present study, these conclusions can be drawn:When analyzing stress distribution in the implants with a frontal load (force applied to both incisors), the stress distribution in analyzed implants was similar, and the equivalent stress did not show many differences between the two variants.When analyzing the stress distribution in the implants with a distal load (force applied to the first premolar), the differences in stress distribution between the unsplinted and splinted variant were not remarkable. The load was transferred right into the body of the implant positioned directly under the first premolar.When analyzing stress distribution in the implants with a distal load (force applied to both molars), the splinted variant transferred the loads with less flexion, whereas in the unsplinted variant, most of the load was localized to the outermost implant.When analyzing the stress distribution in the maxilla, localized peaks of compressive stress were created in the unsplinted variant. This effect was more pronounced in the distal region, where loading was applied to the premolar and molars of the denture.When analyzing the stress distribution in the maxilla in the splinted variant, the bar prevented excessive implant displacement. The compressive stresses are noticeably smaller in all regions where the load was applied.

## Figures and Tables

**Figure 1 materials-16-05248-f001:**
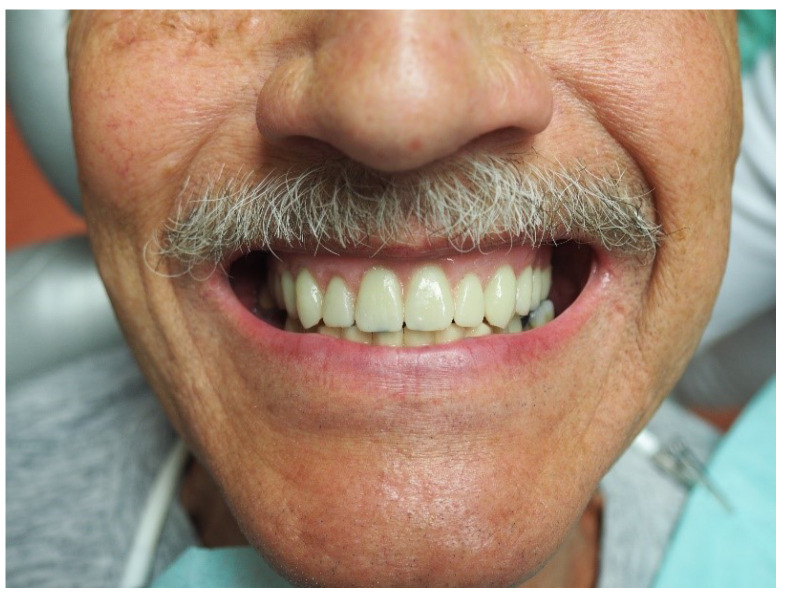
Wax-up try-in.

**Figure 2 materials-16-05248-f002:**
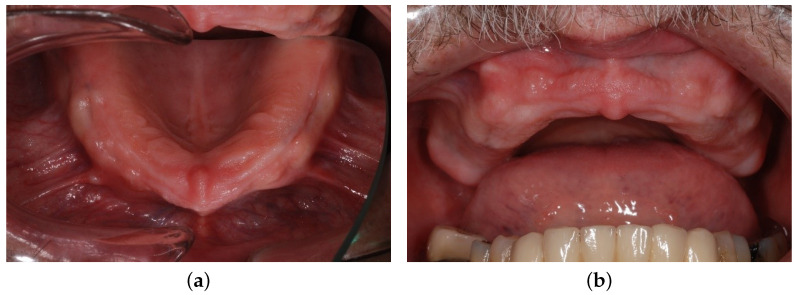
The residual alveolar ridge: occlusal view (**a**) and the front view (**b**).

**Figure 3 materials-16-05248-f003:**
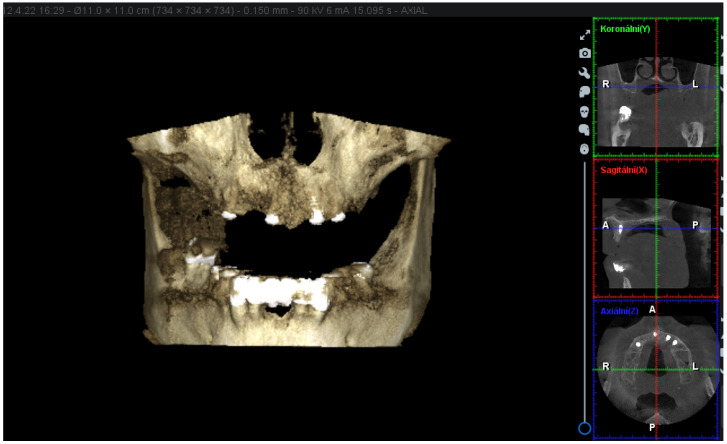
The 3D X-ray scans obtained with Planmeca ProMax${}^{®}$ 3D Classic. R—right, L—left, A—anterior, P—posterior, X—sagittal, Y—coronal, Z—axial.

**Figure 4 materials-16-05248-f004:**
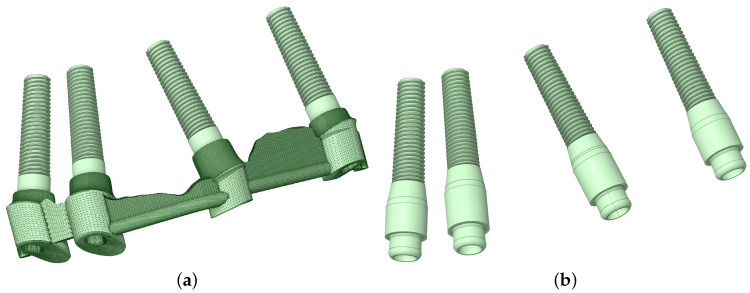
Splinted, bar-supported variant (**a**) and unsplinted, locator-supported variant (**b**).

**Figure 5 materials-16-05248-f005:**
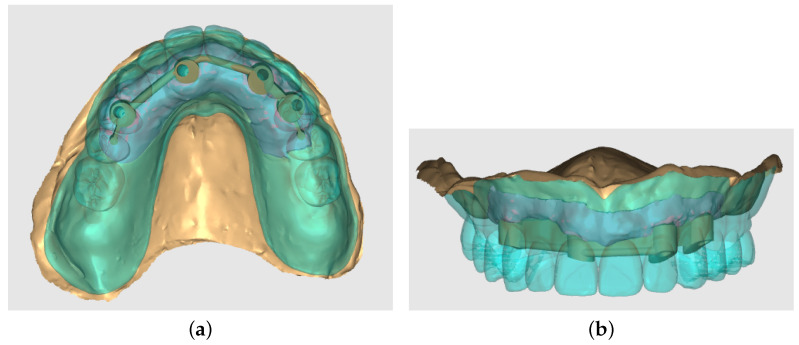
Denture and splinted variant modeled in Exocad Dental CAD program, occlusal view (**a**) and front view (**b**).

**Figure 6 materials-16-05248-f006:**
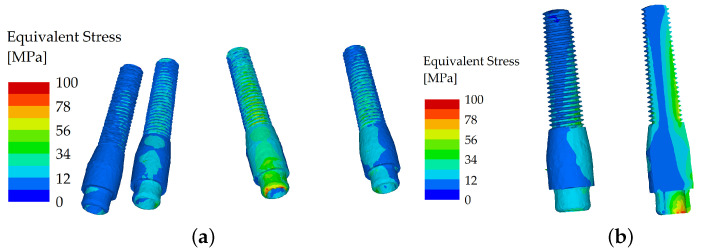
Stress distribution in the dental implants: the load applied in the frontal region (both incisors), unsplinted variant (**a**); and their cross section (**b**).

**Figure 7 materials-16-05248-f007:**
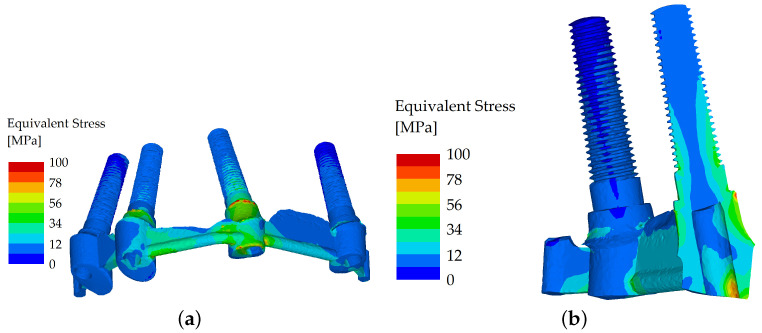
Stress distribution in the dental implants: the load applied in the frontal region (both incisors), splinted variant (**a**); and their cross section (**b**).

**Figure 8 materials-16-05248-f008:**
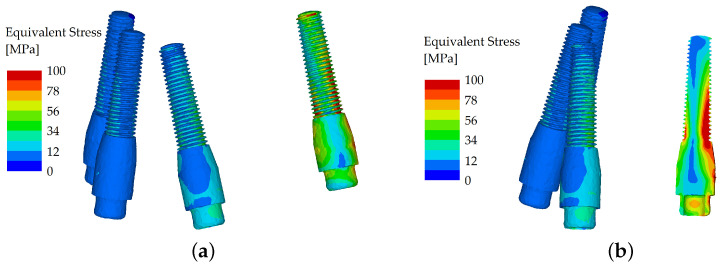
Stress distribution in the dental implants: the load applied in the distal region (molars), unsplinted variant (**a**); and their cross section (**b**).

**Figure 9 materials-16-05248-f009:**
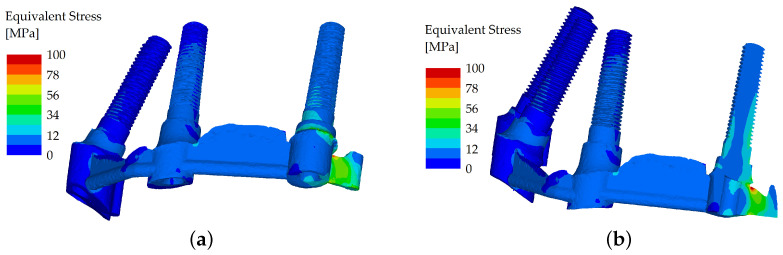
Stress distribution in the dental implants: the load applied in the distal region (molar), splinted variant (**a**); and their cross section (**b**).

**Figure 10 materials-16-05248-f010:**
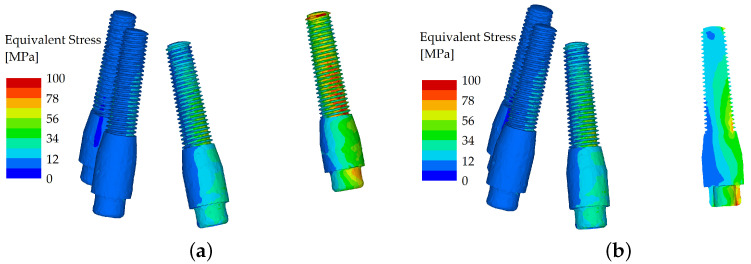
Stress distribution in the dental implants: the load applied in the distal region (premolar), unsplinted variant (**a**); and their cross section (**b**).

**Figure 11 materials-16-05248-f011:**
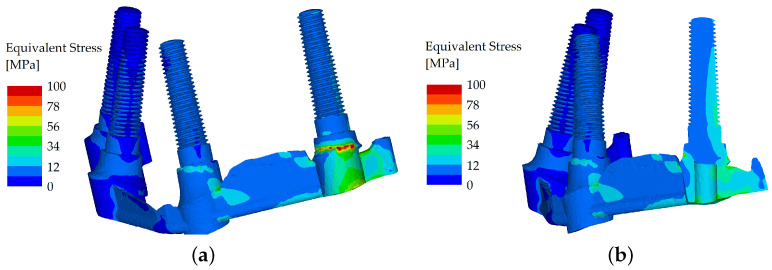
Stress distribution in the dental implants: the load applied in the distal region (premolar), splinted variant (**a**); and their cross section (**b**).

**Figure 12 materials-16-05248-f012:**
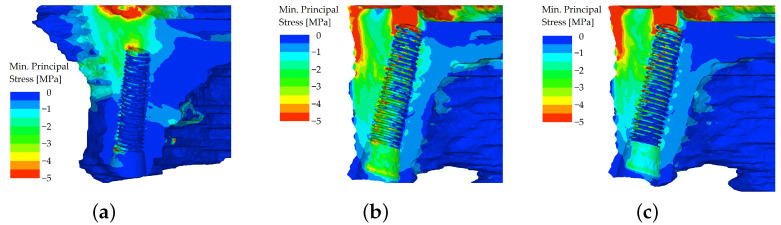
Minimum principal stress distributions in bone in the maxilla. The load is applied in the frontal region (both incisors), unsplinted variant (**a**). The load is applied in the distal region on the first premolar (**b**). The load is applied in the distal region on both molars (**c**).

**Figure 13 materials-16-05248-f013:**
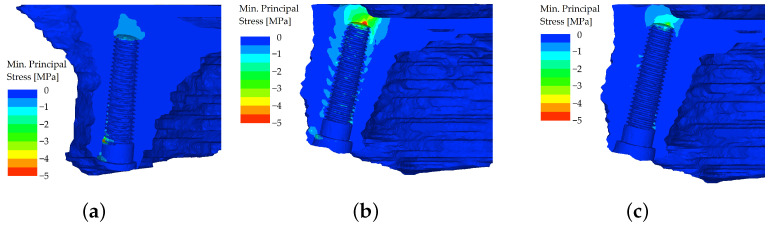
Minimum principal stress distributions in bone in the maxilla. The load is applied in the frontal region (both incisors), splinted variant (**a**). The load is applied in the distal region on the first premolar (**b**). The load is applied in the distal region on both molars (**c**).

**Table 1 materials-16-05248-t001:** Material properties of different components used in the study. Values provided by the manufacturer.

Material	Young Modulus [MPa]	Poisson’s Ratio	References
Dental implants, locators, bar superstructure	108,854	0.28	[21]
Cortical and cancellous bone	QCT/FEA	0.4	[22]
Overdenture (PEEK)	3727	0.4	[23]

**Table 2 materials-16-05248-t002:** The total number of nodes, shells, and solids of both models.

	Splinted Model	Unsplinted Model
No. of Nodes	897,377	739,654
No. of Shells	358,648	290,490
No. of Solids	4,923,388	4,038,705

## Data Availability

Not applicable.

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
