# Peer review of "Biomechanical Analysis of Palateless Splinted and Unsplinted Maxillary Implant-Supported Overdentures: A Three-Dimensional Finite Element Analysis"

_materials, 2023, doi:10.3390/ma16155248_

Round 1

Reviewer 1 Report

The objective of this study was to compare the distribution of stress in the maxillary bone, dental implants, and prosthetic components supporting implant-supported maxillary overdentures 2 with partial palatal coverage, in both splinted and unsplinted designs by a 3D finite element analysis. It is not appropriate for publication in Materials. I recommend the author submit it to other related journals.

Author Response

Dear reviewer, 

thank you for the time to assess our article. As there are no points to respond to, besides your suggestion to publish in other related journals than Materials I can not respond differently. 

Kindest regards

Reviewer 2 Report

The scientific paper "Biomechanical Analysis of Palatless Splinted and Unsplinted Maxillary Implant-supported Overdenture: A 3D Finite Element Analysis” analysed the distribution of stress in the maxillary bone, dental implants, and prosthetic components supporting implant-supported maxillary overdentures with partial palatal coverage, in both splinted and unsplinted designs. The manuscript presents a thorough FEA of the stress distribution in both splinted and unsplinted implant-supported overdentures by employing a comprehensive approach, analyzing stress distribution in the dental implants, as well as in the maxilla for different variants and loading conditions.

While the paper is generally well-conducted, there are several areas that should be improved upon. I can make the following considerations as the Authors may improve their work by following these suggestions:

The authors should provide a more thorough discussion of the limitations of their study. This would give readers a clearer understanding of the context in which the results should be interpreted. For example, why was the study conducted using only splinted and unsplinted designs? Why only one case study? Is this a preliminary research then?

Have authors taken into consideration patient-specific factors such as age, bone quality, and individual anatomy, which could affect the stress distribution in implant-supported overdentures? This should be added and discussed.

The study appears to focus on immediate stress distribution, but what about long-term effects, such as bone remodeling, which could be impacted by the stress distribution?

Other minor suggestions:

In section “Materials and methods” Figure 1 should be referenced first in text above the actual figure, not below it.

For figure 2, a) and b) should be also mentioned in the text above that figure. They should also be named slightly different, as they repeat under Figure 2.

In Figure 5 remove the following text from b) as it repeats “denture and splinted variant modeled in Exocad Dental CAD program” unnecessarily. Same thing applies for Figures 6, 7, 8, 9, 10, 11. Text repeats unnecessarily.

Figures 12 and 13 – correct the round brackets for b) and c). Also, reduce and summarize the text explaining those two figures, it repeats too much.

Round 2

Reviewer 1 Report

In the former report i give a reject decision because i think that the topic of this paper is not appropriate for Materials.  If the editor think it is ok, it can be accepted for publication. 

Reviewer 2 Report

I can see that Authors did acknowledge some of my remarks for their manuscript and improved it.
Good luck in future work!